# ERICT: Enhancing Robustness by Identifying Concept Tokens in Zero-Shot Vision Language Models

**Xinpeng Dong** [* 1]  **Min Zhang** [* 2]  **Didi Zhu** [1]  **Junjian Ye** [3]  **Keli Zhang** [3]  **Aimin Zhou** [2]  **Fei Wu** [1]  **Kun Kuang** [† 1]

## Abstract

Pre-trained vision-language models (VLMs) have revolutionized the field of machine learning, demonstrating exceptional performance across a wide range of tasks. However, their robustness remains vulnerable to the spurious-correlation problem. Existing works often involve fine-tuning the model with labeled data or relying on large language models (LLMs) to generate more complex prompts. Although effective to some extent, these methods introduce new challenges, including additional computational costs and dependence on the quality of prompts without fully utilizing the vision modality. To address these limitations, we propose a novel method named **ERICT** to **E**nhance model **R**obustness by **I**dentifying **C**oncept **T**okens. ERICT mitigates spurious correlation directly in the inference stage and comprises two key steps: (1) Identify concept tokens capturing invariant features through auxiliary prompts to generate a token-level mask. (2) Apply the mask to the attention weights of the CLS token in the vision encoder to help the model focus on the relevant image region. Extensive experiments show that ERICT significantly improves the overall performance, including that of the worst group, and achieves new state-of-the-art results.

## 1. Introduction

Vision language models (VLMs) have achieved remarkable success across various multimodal downstream tasks (Radford et al., 2021; Dehdashtian et al., 2024; Lin et al., 2024; An et al., 2023; Ge et al., 2023). These models are typically built on contrastive language-image pretraining

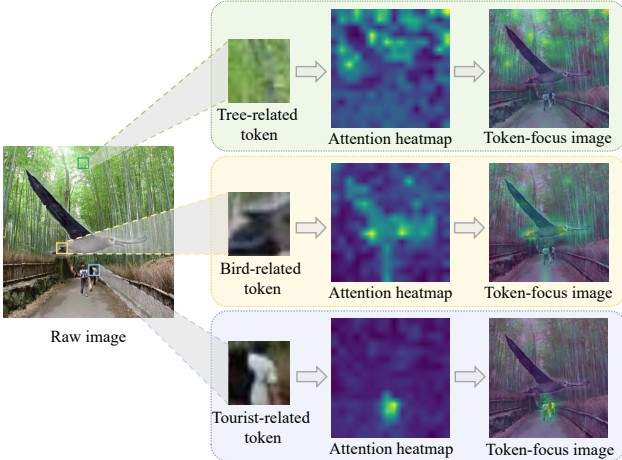

Figure 1: Token attention visualization. We selected three tokens with distinct semantics from the original image and visualized their attention heatmaps, highlighting their corresponding regions in the image. The experiment reveals that the attention of tokens effectively captures semantic information related to specific areas within the image.

(CLIP) (Radford et al., 2021) or similar strategies, leveraging vast amounts of web-scraped data for pretraining. By mapping images and text into a unified representation space, VLMs effectively break the modality barrier between the two. These models can infer the most probable answers by comparing similarities between text and images, demonstrating exceptional zero-shot generalization capabilities. The success of VLMs has introduced a groundbreaking paradigm for modern vision-language model research.

While VLMs have introduced a groundbreaking research paradigm, recent studies (Izmailov et al., 2022; Zheng et al., 2024; Ye et al., 2024a) have revealed their vulnerability to spurious correlations (Sagawa et al., 2019). Specifically, these models may rely on spurious features (*e.g.*, backgrounds) for predictions, leading to performance degradation on samples where spurious information changes. In recent years, a few studies have attempted to address the issue of spurious correlations in VLMs (Yang et al., 2023; Dehdashtian et al., 2024; Zhang & Ré, 2022; Phan et al., 2024). These approaches mitigate spurious correlation by fine-tuning pre-trained VLMs using group-labeled data, sig-

*Equal contribution  [1]Department of Computer Science and Technology, Zhejiang University, Hangzhou, China [2]East China Normal University [3]Huawei Noah's Ark Lab. Correspondence to: Kun Kuang <kunkuang@zju.edu.cn>.

*Proceedings of the 42nd International Conference on Machine Learning*, Vancouver, Canada. PMLR 267, 2025. Copyright 2025 by the author(s).

nificantly improving robustness. However, obtaining group labels typically requires expert human knowledge and fine-tuning introduces significant computational costs. To address the limitations of fine-tuning VLMs, some methods have been proposed to mitigate spurious problems during the inference phase, without relying on additional training or group labels (An et al., 2023; Chuang et al., 2023; Adila et al., 2024). These methods design more reasonable prompts or use large language models (LLMs) to generate additional spurious attributes, applying linear projections to debias visual embeddings. However, these methods either mainly focus on the text modality failing to utilize the information from the vision modality, or the attributes generated by LLMs are inherently unreliable, with linear projection potentially distorting the distribution of visual embeddings.

To achieve better modality synergy which effectively leverages information from both text and vision modalities, we draw inspiration from patch tokens within the vision encoder and are surprised to find that, although these tokens were not explicitly optimized during the pre-training phase, their representations can capture semantic information corresponding to the relevant region in the image. We visualize the attention weight of different tokens in Figure 1. We can observe that the representations of different image tokens focus on the regions of the image that are relevant to themselves. For example, in Figure 1, the token in the last line is related to the concept of tourist, and its attention is concentrated on tourists in the image. Based on this finding, we leverage text prompts to identify which tokens contain task-relevant features and use these features to debias. Unlike methods that focus on a single modality, we incorporate text prompts to guide the vision encoder to focus on specific tokens and get better vision embedding which preserve alignment across text and image modalities.

In this paper, we propose ERICT which enhances the model's robustness during the inference phase **without relying on training, assistance of LLMs or group labels.** For an image, we first use an auxiliary prompt to get the score matrix of all tokens and perform a sparsification operation to generate a token-level mask that reserves the tokens containing task-relevant features. Then we apply the mask to the attention weights of the CLS token in the vision encoder during the inference stage, which can effectively help the model focus on the relevant image region. Furthermore, to address the challenge of acquiring auxiliary prompt words, we propose the ERICT-C. The detailed implementation of the method is provided in Section 4. We conduct extensive experiments on spurious correlation datasets. The results and visualization figures show that our approach effectively mitigates the spurious correlation.

We summarize our contributions as follows: (1) We propose a new approach to mitigate spurious correlation in VLMs

zero-shot inference phase by identifying invariant vision tokens without relying on the assistance of LLMs or group labels. (2) We present a theorem that explains why our approach can effectively handle spurious correlation. (3) We conduct extensive experiments and validate the superiority of our methods via quantitative and visualization results.

## 2. Related Work

**Mitigating spurious correlation in VLMs.** There are many methods try to mitigate spurious correlation in VLMs. These methods can be categorized into two approaches: fine-tuning and zero-shot generalization. Most of them focus on fine-tuning using training data with group label (Yang et al., 2023; Varma et al., 2024; Kim et al., 2023; Eastwood et al., 2024; Zhang et al., 2023; 2024). For example, FairerCLIP (Dehdashtian et al., 2024) appends two adapters and employs the Hilbert Schmidt Independence Criterion (HSIC) to learn spurious features and invariant features separately. CPT (Phan et al., 2024) minimizes the entropy of loss distribution across different groups combining prompt tuning to mitigate spurious correlation. These methods require data with group information for training. However, such data is difficult to obtain and typically requires human expert knowledge for annotation.

Due to the superior performance and convenience of VLMS zero-shot generalization, some researchers focus on mitigating spurious associations during the inference phase without relying on training. Hierarchy-CLIP (Ge et al., 2023) aims to enhance the robustness of prompts by focusing on the semantic parents and children of label categories. PerceptionCLIP (An et al., 2023) utilizes prior knowledge to identify spurious features and then employs these identified features to enhance prompts. Chuang et al. (2023) train a projection using the constructed text pairs and project the text embeddings into a space orthogonal to the spurious attribute space. ROBOSHOT (Adila et al., 2024) generates insight for spurious features with LLMs and applies a linear projection to map image embeddings to a neutralization hyperplane for spurious features. These works either only consider the textual modality without considering the vision modality, or rely on the help of LLMs. Our method combines information from two modalities and does not require any training data or LLMs assistance.

**Difference from token pruning.** Currently, many approaches solving vision-language model tasks emphasize manipulating tokens and attention weights, and token pruning is one of the most popular applications. The main goal of existing token pruning methods is to improve model efficiency (He et al., 2021; Tang et al., 2023; Wang et al., 2023; Ye et al., 2024b). Generally, token pruning can be achieved through two primary methods: attention mask pruning (Rao et al., 2021) and activation mask pruning (Kim et al., 2022).

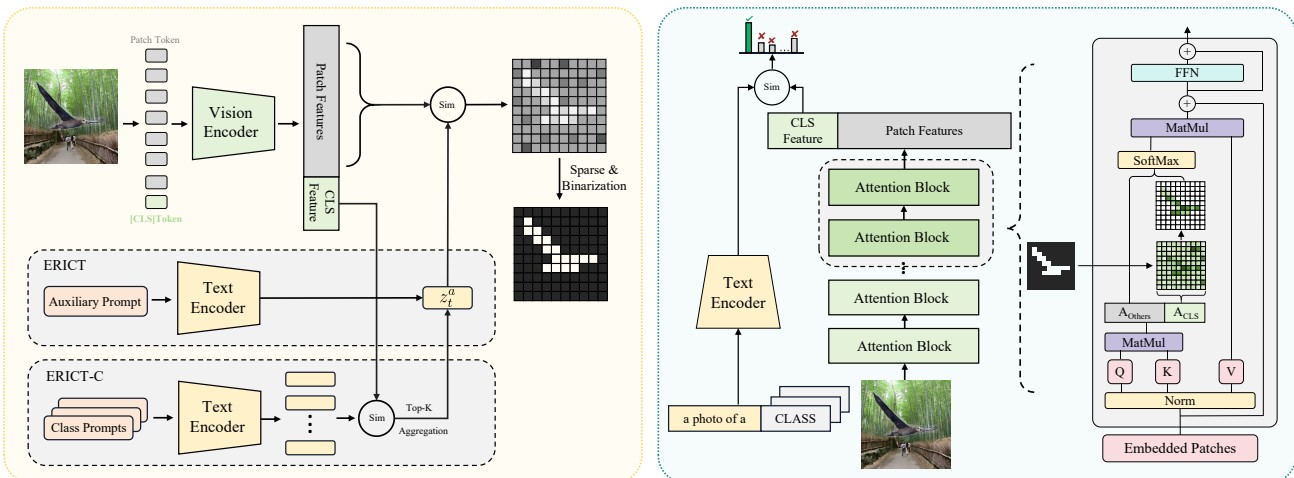

Step 1: Identify invariant tokens                    Step 2: Mitigate during inference

Figure 2: Overall framework. Our framework mainly includes two steps: In Step 1, we construct an auxiliary embedding $z_t^a$ to identify tokens containing invariant information, obtaining a token-level mask. In Step 2, we apply the mask within the attention mechanism of the vision encoder, making tokens containing spurious information invisible to the [CLS] token.

Our method is partly inspired by attention mask pruning, but different from it. Token pruning involves removing tokens that carry few information or are redundant during the attention computation process. In contrast, our approach aims to direct the model's attention to specific regions of the image. In terms of method implementation, we don't remove the tokens completely and all tokens undergo the full attention computation process. We unilaterally reduce the attention of the CLS token on certain tokens. In this way, we enable the model to produce representations with more invariant features rather than spurious features.

## 3. Preliminary

In this section, we first give the problem definition, and then provide a detailed description of vision-language models.

**Problem definition.** This paper primarily focuses on addressing the problem of group robustness (Sagawa et al., 2019). Specifically, let $x \in X$ represent the input image, $y \in Y$ the target label, and $a \in A$ the spurious feature. A dataset with spurious correlations $(y, a)$ is annotated with a group label $g \in G$, where $G := Y \times A$ denotes the set of all possible combinations of class labels and spurious attributes. To mitigate the impact of spurious correlations on predictions, our method follows the basic goal of existing works (Liu et al., 2021; Asgari et al., 2022; Le et al., 2024; Arefin et al., 2024), aiming to improve the accuracy of the worst group and narrow the gap between the worst group performance and average performance.

**Vision-language models (VLMs).** VLMs contain two types of encoders: a vision encoder $f_v^\theta$, parameterized by $\theta$ and a textual encoder $f_t^\phi$, parameterized by $\phi$. For simplicity, we omit $\theta$ and $\phi$ in the following. The vision encoder maps

the input image $x_v \in \mathbb{R}^{3 \times W \times H}$ to a $h$-dimensional vision embedding $z_v = f_v(x_v) \in \mathbb{R}^h$, where $W$ and $H$ denote the width and height of the image. The textual encoder processes the corresponding textual information $x_t$ to generate its embedding $z_t = f_t(x_t)$, where $z_t \in \mathbb{R}^{L \times B}$ with $L$ representing the text length and $B$ representing the dimension. Taking the CLIP model (Radford et al., 2021) as an example, a representative approach in VLMs, the specific form of its optimization loss function is presented as follows:

$$\mathcal{L}_{CLIP} = -\sum_{i=1}^{N} \log \frac{\exp\big(d(z_v^i, z_t^i)/\tau\big)}{\sum_{j=1}^{N} \exp\big(d(z_v^i, z_t^j)/\tau\big)}, \quad (1)$$

where $d(\cdot)$ denotes the cosine similarity. $\tau$ is a learnable temperature parameter. $N$ means the number of samples.

Specifically, taking the classification downstream task as an example, in the zero-shot inference stage, hard prompts are used as text to generate a zero-shot classifier, such as "A photo of {CLASS}". After obtaining vision and textual embeddings, CLIP is classified by calculating their similarity:

$$\hat{y} = \operatorname{argmax}_{i \in N_c} d(z_v, z_t^i), \quad (2)$$

where $\hat{y}$ represents the class label predicted by model and $N_c$ represents the number of classes.

## 4. Methodology

In this section, we propose ERICT and ERICT-C to mitigate the spurious correlation during zero-shot inference of VLMs. Our methods don't require any training data or group labels, making it both efficient and broadly applicable. There are two main steps: (1) identify invariant vision tokens in Section 4.1 and (2) mitigate spurious correlation by leveraging the discovered invariant information in Section 4.2.

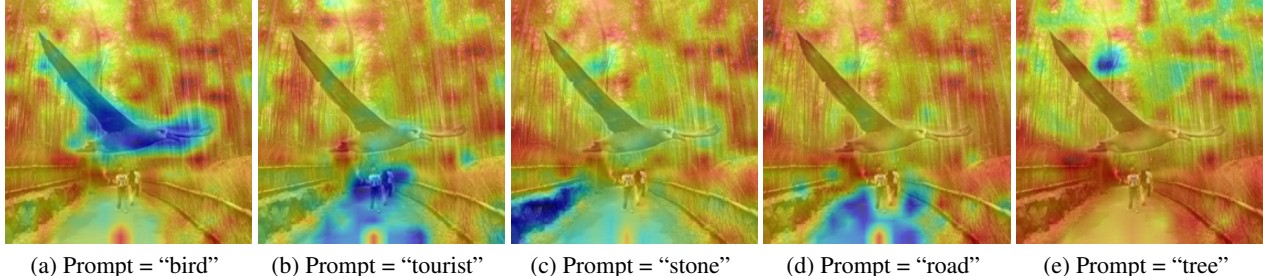

(a) Prompt = "bird"    (b) Prompt = "tourist"    (c) Prompt = "stone"    (d) Prompt = "road"    (e) Prompt = "tree"

Figure 3: Score maps with different prompts. We use different prompts to compute score matrices with the representations of all tokens. The heatmaps show that scores of relevant tokens are significantly lower than other tokens.

### 4.1. Step 1: Identify Invariant Vision Tokens

For every image, we try to identify which tokens focus on invariant information and which tokens focus on spurious information. It is achieved based on our finding.

**Finding.** As mentioned in Section 1, in vanilla CLIP inference process, the image patch token representations are discarded, even though these representations contain rich semantic information. We discovered that this semantic information can be captured with specific prompts. Specifically, we calculate the similarity score matrix between a prompt and the representations of all the image tokens. Different from the CLS token, we observed that tokens whose semantics align more closely with the prompt tend to have **lower** similarity scores, as shown in Figures 3. Based on this discovery, the difference in scores allows us to identify the tokens focusing on invariant features within the image, and we achieve this via a token-level mask.

The overall framework is shown in Figure 2. For ERICT, we use an auxiliary prompt $x_t^a$ for every task and get the auxiliary text feature by the text encoder. The specific method for selecting auxiliary prompts will be introduced later.

Then, given an image token embedding sequence $\mathbf{X} = [\mathbf{x}_{cls}, \mathbf{x}_1, \ldots, \mathbf{x}_p]^T$ for attention blocks in Vision Transformer(ViT), where $x_{cls}$ represents the added global class token, $\{x_i | i = 1, 2, \ldots, p\}$ denote local patch tokens and p represents the number of patch, we compute the score matrix $S \in \mathbb{R}^p$ between the final token embeddings $\mathbf{X}_{patch} = \mathbf{X} \setminus \{\mathbf{x}_{cls}\}$ and the auxiliary prompt:

$$S = d(\mathbf{X}_{patch}, f_t(x_t^a)). \tag{3}$$

According to the obtained score matrix, we generate the corresponding mask $M$. A common method is to set a percentage threshold, but the proportion of related attributes varies in different images. So we use a method similar to SparseMax (Martins & Astudillo, 2016) to generate a specific mask for every image.

Let $S' = \{S_i'\}_{i=1}^p = Sort(-1 \times S)$, which means $S_1' \geq S_2' \geq \cdots \geq S_p'$. we firstly find the max index $k$ that satisfies

the following conditions:

$$k = \max\{k \in [p] \mid 1 + kS_k'/\tau > \textstyle\sum_{j \leq k} S_j'/\tau\}, \tag{4}$$

where $\tau$ is the temperature parameter which we will analyze in the ablation study. Then we calculate the threshold $l$:

$$l = \frac{\left(\sum_{j \leq k} S_j'/\tau\right) - 1}{k}. \tag{5}$$

The mask $M$ is determined based on the relationship between the score and the threshold.

$$M_i = \begin{cases} 0 & \text{if } S_i' \leq l \\ 1 & \text{if } S_i' > l \end{cases} \tag{6}$$

**Auxiliary embedding.** The auxiliary embedding $z_t^a$ is pretty important in this process. For ERICT, $z_t^a = f_t(x_t^a)$. Ideally, the auxiliary prompt $x_t^a$ should be the ground-truth category name or description of relevant features corresponding to the image, but it is not feasible during the zero-shot inference stage. So we adopt the superclass of the task category with a template "{} in photo". For example, on the CelebA dataset, we use "hair in photo" as an auxiliary prompt. Furthermore, when it is difficult to determine the superclass, we propose ERICT-C whose auxiliary embedding can be obtained through aggregating class prompt embeddings. For datasets with a small number of categories (eg., Waterbirds), it is sufficient to directly aggregate the representations of all categories.

$$z_t^a = \frac{1}{N_c} \sum_{i \in N_c} f_t(x_t^i), \tag{7}$$

where $C$ denotes the set of classes and $z_t^a$ represents embedding of auxiliary prompt, which corresponds to $f_t(x_t^a)$ in the previous equation.

When the dataset contains a large number of class (e.g., ImageNet), ERICT-C adopt a top-K strategy. Specifically, in Step 1, we first compute the model's top-K outputs, and then merge the corresponding k textual representations. Step 1 already involves a inference, so the top-k strategy does not require an additional inference.

$$z_t^a = \frac{1}{K} \sum_{k \in K} f_t(x_t^k). \tag{8}$$

We present the results of ERICT-C on the ImageNet series datasets in the Appendix D.1.

### 4.2. Step 2: Mitigate Spurious During Inference

In this step we perform the actual debiasing and inference. Existing works (Chen et al., 2022; Li et al., 2023a; Lin et al., 2024) have demonstrated that attention weights can capture the relationships between different patches, thereby reflecting the regions that the model focuses on.

Based on the token-level mask obtained in Step 1, we enforce the model to focus on the regions containing invariant information. Specifically, we apply the mask to the attention weight of CLS token in vision encoder. In this case, the CLS token's perception of certain tokens is blocked, and these tokens correspond to the parts containing spurious information. Only the tokens containing invariant information are retained for further perception. The attention process with mask becomes as follow:

$$\mathbb{A}(x_{cls}, :) = \text{softmax}(M \cdot \frac{Q_{cls}K^T}{\sqrt{d_k}}), \qquad (9)$$

$$\mathbb{A}(x_{patches}, :) = \text{softmax}(\frac{Q_{patches}K^T}{\sqrt{d_k}}), \qquad (10)$$

$$\mathbf{X} = \mathbf{X} + \text{Proj}([\mathbb{A}(x_{cls}, :), \mathbb{A}(x_{patches}, :)] \cdot V), \qquad (11)$$

where $Q, K, V$ are query matrix, key matrix, value matrix in attention calculation process respectively and Proj is related projection module in the vision encoder. $d_K$ is the dimension of $K$. $\mathbb{A}$ represents the $Q$-$K$ attention weight.

Then, we construct prompts using a template and class name like vanilla CLIP. Image classification is performed using the similarity between image representation and text representation. For all $C$ classes:

$$\hat{y} = argmax_{i \in N_c} d(f_v(x_v, M), f_t(x_t^i)), \qquad (12)$$

where $x_t^i$ is a single class prompt simply constructed with "a photo of a {CLASS}" template.

## 5. Theory Analysis

To better understand why the binary mask can effectively mitigate spurious correlation, in this section, we present an error probability bound of the model under the masking condition and demonstrate that using a binary mask can effectively reduce the error bound. Firstly, we define the task in the context of vision-language settings:

**Definition 5.1.** Consider a classification task with $n$ pairs of $\{x_v^i, x_t^i\}_{i=1}^n$. Every image $x_v^i$ is generated from the underlying latent factor $Z = [Z_{inv}, Z_{spu}]$ which is composed of the invariant factor $Z_{inv} \sim \mathcal{N}(y, \sigma_{inv}^2)$ and spurious factor $Z_{spu} \sim \mathcal{N}(a, \sigma_{spu}^2)$. Prompt $x_t^i$ is generated from the

ground truth $y_i$. The ground truth $y$ is uniformly drawn from $\{-1, 1\}$, and the spurious attribute $a$ drawn from $\{-1, 1\}$ with the probability $p_{spu} = Pr(a = y)$, where $1 > p_{spu} > 1/2$. The variance $\sigma^2$ denotes the variation of the corresponding variable.

For the architecture, we consider two simple linear encoders $f_v : \mathbb{R}^{d_I} \to \mathbb{R}^h$ for the image modality and $f_t : \mathbb{R}^{d_T} \to \mathbb{R}^h$ for the text modality, implemented as $f_v(x_v) = \mathbf{W_I}x_v$ and $f_t(x_t) = \mathbf{W_T}x_t$ with $\mathbf{W_I} \in \mathbb{R}^{h \times d_I}$ and $\mathbf{W_T} \in \mathbb{R}^{h \times d_T}$ respectively. In the vanilla CLIP zero-shot inference phase, the decision is made by computing a similarity score, which takes the following form:

$$x_v^T \mathbf{W}_I^* \mathbf{W}_T^* x_t^{\hat{y}} \qquad (13)$$

**Theorem 5.2.** *Given a task defined in 5.1 and an over-parameterized CLIP model where $n = \omega(1), d_I = \Omega(n)$ and $d_T = \Omega(n)$. A distinguishable mask $M \sim Bernoulli(p_{mask})$ aims to remove spurious components from the vision modality. If the gap between the variances of the core and spurious features is significant: $\sigma_{inv} = \Theta(1), \sigma_{inv} \geq 1$ and $\sigma_{spu} = O(\frac{1}{\sqrt{\log n}})$. Then with a high probability of at least $1 - O(\frac{1}{poly(n)}) = 1 - o(1)$, the CLIP model achieves a large error in zero-shot accuracy in the out-of-distribution test data where $a \neq y$:*

$$\Phi(\kappa) - o(1) \leq Err_{a \neq y} \leq \Phi(\kappa) + o(1),$$

*where $k = \frac{p_{mask}(2p_{spu}-1)-1-\sigma_{inv}}{\sqrt{(1+\sigma_{inv}^2)^2\sigma_{inv}^2 + (2p_{spu}-1)^2 p_{mask}^2 \sigma_{spu}^2}}$ and $\Phi$ denotes the CDF of a standard normal distribution.*

Notice that $0 \leq p_{mask} \leq 1$ and $1 - p_{mask}$ is the actual mask ratio ( $p_{mask} = 1$ represents the vanilla error bound without mask ) . As the proportion of the masked spurious components increases, the model error bound gradually decreases. We defer the proof of Theorem 5.2 to Appendix A, where a detailed derivation is provided.

## 6. Experiments

### 6.1. Experiment Setting

**Datasets.** We evaluate our approach on three widely used spurious correlation datasets, including Waterbirds (Sagawa et al., 2019), CelebA (Liu et al., 2015), and Urbancars (Li et al., 2023b). For Waterbirds and CelebA, we follow the setting of previous works (Sarridis et al., 2024; Yang et al., 2024; You et al., 2024). Waterbirds dataset defines a task to identify whether a bird is a land bird or water bird, and the spurious attribute is background. CelebA dataset defines a task to identify whether a person's hair is blonde or not and the spurious attribute is gender. Urbancars is an artificially generated dataset to identify car types, and the spurious attribute are background and co-occurrence objects.

Table 1: Experiments on Waterbirds. We highlight the best results in **bold**, the second-best results with an underline.

| | ViT-B/32 | | | ViT-B/16 | | | ViT-L/14 | | |
|---|---|---|---|---|---|---|---|---|---|
| | WG (↑) | AVG (↑) | Gap (↓) | WG (↑) | AVG (↑) | Gap (↓) | WG (↑) | AVG (↑) | Gap (↓) |
| ZSCLIP | 39.33 | 67.34 | 28.01 | 22.74 | 79.23 | 56.49 | 35.20 | 84.17 | 48.97 |
| Group Prompt | 64.17 | 79.91 | 15.74 | 16.82 | 81.51 | 64.69 | 29.43 | 85.22 | 55.79 |
| PerceptionCLIP | 66.07 | 89.8 | 23.73 | 16.07 | 82.98 | 66.91 | 44.94 | 86.44 | 41.51 |
| PerceptionCLIP+ | 60.33 | 78.6 | 18.27 | 41.07 | 85.8 | 44.73 | 61.12 | 87.74 | 26.62 |
| ROBOSHOT | 54.4 | 82.0 | 27.6 | - | - | - | 45.2 | 79.9 | 34.7 |
| ERICT | **71.44** | 78.67 | **7.23** | 56.23 | 83.81 | 27.58 | 56.39 | 88.66 | 32.27 |
| ERICT-C | 57.2 | 72.52 | 15.23 | **57.32** | 82.46 | **25.14** | **62.31** | 88.09 | **25.78** |

Table 2: Experiments on CelebA. We highlight the best results in **bold**, the second-best results with an underline.

| | ViT-B/32 | | | ViT-B/16 | | | ViT-L/14 | | |
|---|---|---|---|---|---|---|---|---|---|
| | WG (↑) | AVG (↑) | Gap (↓) | WG (↑) | AVG (↑) | Gap (↓) | WG (↑) | AVG (↑) | Gap (↓) |
| ZSCLIP | 61.11 | 90.48 | 29.37 | 68.88 | 88.35 | 19.47 | 72.77 | 87.68 | 14.91 |
| Group Prompt | 45.32 | 88.46 | 43.14 | 67.09 | 86.13 | 19.04 | 75.00 | 89.09 | 14.09 |
| PerceptionCLIP | 76.70 | 79.89 | 3.19 | 65.13 | 75.27 | 10.14 | 74.31 | 80.30 | 5.99 |
| PerceptionCLIP+ | 75.94 | 82.02 | 6.08 | 69.18 | 77.17 | 7.99 | 80.84 | 83.04 | 2.20 |
| ROBOSHOT | 80.5 | 84.8 | 4.3 | - | - | - | 82.6 | 85.5 | 2.9 |
| ERICT | 83.46 | 86.53 | 3.07 | 80.00 | 86.49 | 6.49 | 85.00 | 86.43 | 1.43 |
| ERICT-C | **83.62** | 86.25 | **2.63** | **81.98** | 84.30 | **2.32** | **85.44** | 86.53 | **1.09** |

**Baselines.** We compare our method with state-of-the-art methods, including vanilla zero-shot CLIP (ZSCLIP), ZSCLIP with group information (Group Prompt), PerceptionCLIP (An et al., 2023) and ROBOSHOT (Adila et al., 2024). Group Prompt assumes access to spurious attributes and includes them in the label prompts. For instance, the label prompts of Waterbirds dataset become "a photo of waterbird with water background, a photo of waterbird with land background, a photo of landbird with water background, a photo of landbird with land background". For PerceptionCLIP, it uses a variety of different combinations of assists. We show all of them, PerceptionCLIP represents using the standard spurious attribute (*e.g.*, gender for CelebA dataset), and PerceptionCLIP+ represents using extended attributes (*e.g.*, gender, age, and race for CelebA dataset).

### 6.2. Main Results

We present results on three widely used backbones, including ViT-B/16, ViT-B/32 and ViT-L/14. Generally, our method has a significant improvement on the worst group performance and narrows the gap between the worst group and the average performance among all backbones.

Table 1 shows the results on the Waterbirds dataset. Our methods perform well across all backbones, especially on the base backbone, where it shows significant improvement compared to baseline methods. On the ViT-B/32 backbone, ERICT achieves a worst-case group accuracy of 71.44%. And on the ViT-B/16 backbone, ERICT-C achieves a worst-case group accuracy of 57.32%, surpassing the current best-performing method by a margin of 16.25 percentage points.

Table 2 presents the results on the CelebA dataset. Our methods perform outstandingly on the CelebA dataset, with ERICT-C and ERICT achieving the best and second best performance across all backbones, respectively. In particular, with the ViT-B/16 backbone, our methods achieve an improvement of more than 10 percentage points on the worst group compared to the previous baseline methods. Additionally, we significantly reduced the gap between the performance of the worst group and the average performance, narrowing the gap to 1.09 on the ViT-L/14 backbone.

We also conduct experiments on the Urbancars dataset in Table 3, which poses a significant challenge for CLIP. Nevertheless, our method remains effective: on the ViT-L/14 backbone, it improves the accuracy of the worst group performance from 12.4% to 49.6%. Importantly, this improvement does not come at the cost of average performance; in fact, we observe a notable enhancement in average accuracy.

Overall, as shown in Tables 1 2 3, ERICT and ERICT-C consistently outperform all other methods across different datasets and backbones, setting a new state-of-the-art performance. However, we observed that ERICT does not show significant improvement in some cases, and ERICT-C outperforms ERICT in many cases. The phenomenon points to the suboptimality impact of the auxiliary prompt, which we will discuss in following sections.

Table 3: Experiments on Urbancars. We highlight the best results in **bold**, the second-best results with an underline.

| | ViT-B/32 | | | ViT-B/16 | | | ViT-L/14 | | |
|---|---|---|---|---|---|---|---|---|---|
| | WG (↑) | AVG (↑) | Gap (↓) | WG (↑) | AVG (↑) | Gap (↓) | WG (↑) | AVG (↑) | Gap (↓) |
| ZSCLIP | 14.80 | 51.60 | 36.80 | 0.00 | 52.00 | 52.00 | 12.40 | 54.00 | 41.60 |
| Group Prompt | 33.20 | 53.50 | 20.30 | 7.60 | 52.70 | 45.10 | 25.20 | 51.30 | 26.10 |
| ERICT | **38.80** | 53.30 | **14.50** | **9.20** | 51.40 | **42.20** | **49.60** | 63.30 | **13.70** |
| ERICT-C | 29.60 | 53.00 | 23.40 | 8.40 | 51.70 | 43.30 | 48.00 | 62.60 | 14.60 |

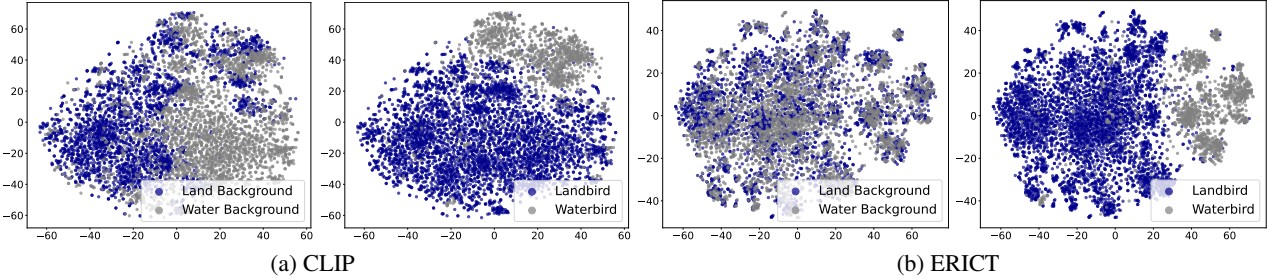

(a) CLIP        (b) ERICT

Figure 4: t-SNE comparison diagram on the Waterbirds dataset. The output representations of the ERICT method in the t-SNE visualization exhibit limited distinction with respect to background information, indicating that the model's decision-making does not rely on background information (*i.e.,* spurious attribute).

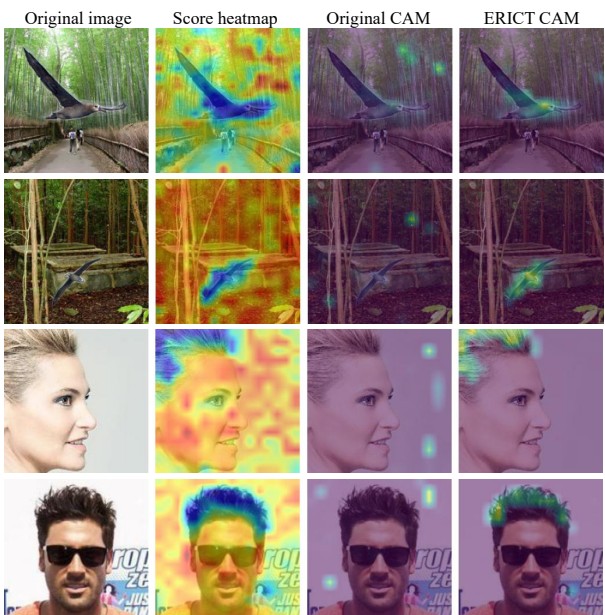

Original image    Score heatmap    Original CAM    ERICT CAM

Figure 5: CAM comparison diagram. We visualized some images from the Waterbirds and CelebA datasets, showing that our approach successfully focuses on the desired region.

## 6.3. Visualization Results

Beyond the quantitative analysis of the datasets, we further provide a detailed demonstration of ERICT's debiasing capability by presenting visualized results.

Considering the widespread use of CAM (Zhou et al., 2016; Jung & Oh, 2021; Vilas et al., 2024) images for interpreting what model focus on, we compared the CAM images of the final layer in the model before and after ERICT application. As illustrated in Figure 5, it can be observed that, in comparison to the vanilla zero-shot CLIP, ERICT effectively guides the model's attention toward the invariant features, thereby mitigating the impact of spurious attributes on inference outcomes. For instance, in the Waterbirds dataset, the vanilla zero-shot CLIP erroneously focuses on background elements, whereas our method successfully redirects attention to the invariant features, specifically the bird.

Considering that our method directly adjusts the attention mechanism, CAM images may not accurately reflect the debiasing capability. Therefore, we also visualize the model's output representations using t-SNE (Van der Maaten & Hinton, 2008), with separate divisions based on ground-truth label and spurious attribute. We present the t-SNE results for the Waterbirds dataset in Figure 4. The vanilla zero-shot CLIP separates the two types of background distinctly in the feature space, indicating that it considers background features as an important discriminative factor. The t-SNE visualization of our method shows that after applying ERICT, the model no longer focuses on the background, which is a spurious attribute. All the images were generated using the default t-SNE parameters from the scikit-learn package.

## 6.4. Ablation Study

In this section, we present the results of several ablation studies aiming to identify key factors that influence the performance of our proposed method. Specifically, we examine the impact of the temperature parameter, the effect of mask placement, and the role of auxiliary prompt words.

**Impact of temperature.** In this section, we investigate the impact of the temperature parameter $\tau$. The temperature parameter controls the sharpness of the similarity score matrix distribution, thereby influencing the mask ratio during the inference phase. A lower temperature value results in a higher mask ratio. The optimal temperature parameter varies depending on specific dataset. In general, the lower the proportion of invariant features in an image, the smaller the temperature parameter. For example, the temperature parameter for the Waterbirds dataset is lower than that for the CelebA dataset, as the proportion of bird features in the Waterbirds dataset is smaller, whereas the proportion of hair features is larger in the CelebA dataset.

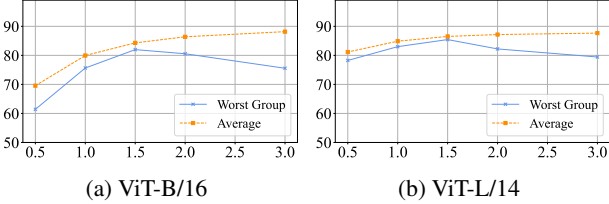

(a) ViT-B/16        (b) ViT-L/14

Figure 6: The impact of different temperature parameters on the CelebA. As the temperature gradually increases, the performance of the worst group exhibits an initial upward trend followed by a subsequent decline.

We illustrate the impact of different temperature parameters in Figure 6. As the temperature parameter decreases, the performance of the worst-performing group initially improves, followed by a decline. This can be explained by the following reasoning: as the mask ratio increases, the model's robustness improves. However, when the mask ratio becomes excessively high, some invariant features are also masked, resulting in a performance degradation. The optimal results are presented in the main experiments.

**Application location of mask.** In this section, we analyze the impact of the applied mask location. Previous studies (Kirichenko et al., 2022; Gandelsman et al., 2023) have emphasized the critical role of the final layers, particularly the last two layers, in enhancing the model's robustness. Therefore, we apply the mask to the last two layers of the vision encoder. To assess its effect, we conduct a comparative experiment where the mask is applied to various locations: the original image, the last 1-3 layers in vision encoder, the latter half of the vision encoder, and all layers in vision encoder. The detailed results are presented in Table 4.

As shown in Table 4, we observe that, compared to the vanilla CLIP, various masking strategies contribute to improved robustness. Among these strategies, applying the mask to the CLS weight of the last two layers achieves the best performance on the worst-performing group. The results presented in our main experiments correspond to those obtained by masking the last two layers.

Table 4: Effect of mask location with ViTL14 backbone on CelebA dataset. In our method, the mask is applied to the CLS weights of the last two layers.

|  | WG ($\uparrow$) | AVG ($\uparrow$) | Gap ($\downarrow$) |
|---|---|---|---|
| ZSCLIP | 72.77 | 87.68 | 14.91 |
| Mask image | 81.67 | 90.30 | 8.63 |
| Last 1 layer | 83.33 | 85.48 | 2.15 |
| Last 2 layers | **85.44** | 86.53 | **1.09** |
| Last 3 layers | 84.44 | 87.37 | 2.93 |
| Last half backbone | 80.00 | 88.61 | 8.61 |
| All layers | 79.44 | 88.18 | 8.74 |
| Token purning | 83.89 | 87.91 | 4.02 |

**Impact of the auxiliary prompt.** In our approach, auxiliary prompt words play a critical role, as they determine the key information that the model focuses on. In this paper, we propose ERICT, which computes auxiliary representations using parent class prompts, and ERICT-C, which employs clustering based on class prompt representations as auxiliary representations. Due to the limitations of model's capabilities, the current recognition stage remains relatively simplistic, capturing primarily basic semantics. The auxiliary information used by ERICT and ERICT-C offers a viable solution, though it is not the optimal one. We hope that future work can address this challenge.

## 7. Conclusion

In this paper, we aim to enhance the robustness of vision language models during zero-shot inference without requiring any additional training or external model assistance. To achieve this, we proposed ERICT and ERICT-C, which identify concept tokens that capture invariant features and leverage this invariant knowledge to effectively alleviate spurious correlation during inference. Through experiments and visualizations, we demonstrate the effectiveness of our approach, which significantly reduces the impact of spurious correlations and achieves notable performance improvements compared to existing baseline methods across multiple datasets and backbones. Ablation studies on various experimental settings explain the impact of temperature parameter and mask location on performance. Additionally, for CLIP-based VLMs, we provided a new theory supporting the effectiveness of masks to enhance robustness, providing strong theoretical support for our method. Although our proposed method demonstrates significant performance, it still has some limitations. The current model relies on specific auxiliary embedding, which don't guarantee the optimality. Future research could explore the following directions: investigating how to obtain more effective auxiliary text representation and optimizing mask strategies.

## Acknowledgements

This work was supported in part by the National Key Research and Development Program of China (2024YFE0203700), National Natural Science Foundation of China (62376243, 62441605), and "Pioneer" and "Leading Goose" R&D Program of Zhejiang (2025C02037). All opinions in this paper are those of the authors and donot necessarily reflect the views of the funding agencies.

## Impact Statement

This paper presents work whose goal is to advance the field of Machine Learning. There are many potential societal consequences of our work, none which we feel must be specifically highlighted here.

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

## A. Proof of Theory

**Definition A.1.** Consider a classification task with $n$ pairs of $\{x_v^i, x_t^i\}_{i=1}^n$. Every image $x_v^i \in \mathbb{R}^{d_I}$ is generated from the underlying latent factor $Z = [Z_{inv}, Z_{spu}]$ which is composed of the invariant factor $Z_{inv} \sim \mathcal{N}(y, \sigma_{inv}^2)$ and spurious factor $Z_{inv} \sim \mathcal{N}(a, \sigma_{spu}^2)$. Text $x_t^i \in \mathbb{R}^{d_T}$ is generated from the ground truth $y_i$. The ground truth $y$ is uniformly drawn from $\{-1, 1\}$, and the spurious attribute drawn from $\{-1, 1\}$ with the probability $Pr(a = y) = p_{spu}$, where $1 > p_{spu} > 1/2$. The variance $\sigma^2$ denotes the variation of the corresponding variable.

With the definition, we can write $Z = \begin{bmatrix} y + \xi_1 \\ a + \xi_2 \end{bmatrix}$ where $\xi_1 \sim \mathcal{N}(0, \sigma_{inv}^2), \xi_2 \sim \mathcal{N}(0, \sigma_{spu}^2)$ are two Gaussian variables in the definition. The $x_v$ is generated via $x_v = \boldsymbol{D}_I Z$, text $x_t$ is generated via $y = \boldsymbol{D}_T \begin{bmatrix} y \\ 0 \end{bmatrix}$, with $\boldsymbol{D}_I \in \mathbb{R}^{d_I \times l}$ and $\boldsymbol{D}_T \in \mathbb{R}^{d_T \times l}$. The matrix $\boldsymbol{D}_I$ and $\boldsymbol{D}_T$ is a matrix with orthonormal columns which can be considered as a dictionary matrix. And the distinguishable mask $M \sim \text{Bernoulli}(p_{mask})$ removes spurious components from the vision modality.

In clip architecture, there are two encoders $f_v : \mathbb{R}^{d_I} \to \mathbb{R}^h$ for the image modality and $f_t : \mathbb{R}^{d_T} \to \mathbb{R}^h$ for the text modality, implemented as $f_v(x_v) = \mathbf{W_I} x_v$ and $f_t(x_t) = \mathbf{W_T} x_t$ with $\mathbf{W_I} \in \mathbb{R}^{h \times d_I}$ and $\mathbf{W_T} \in \mathbb{R}^{h \times d_T}$ respectively. In the vanilla CLIP zero-shot inference phase, the decision is made by computing a similarity score, which takes the following form:

$$x_v^T \boldsymbol{W}_I^* \boldsymbol{W}_T^* x_t^{\hat{y}} \tag{14}$$

Before the formal proof, we firstly introduce a useful lemma:

**Lemma A.2** ((Xue et al., 2024)). *The minimizer of linearized CLIP loss $\boldsymbol{W}_I^{*T} \boldsymbol{W}_T^*$ satisfies the following with a probability of at least $1 - O(\frac{1}{poly(n)})$ such that,*

$$||\boldsymbol{W}_I^{*T} \boldsymbol{W}_T^* - \frac{1}{\rho} \boldsymbol{D}_I \begin{bmatrix} 1 + \sigma_{inv}^2 & 2p_{spu} - 1 \\ 2p_{spu} - 1 & 1 + \sigma_{spu}^2 \end{bmatrix} \boldsymbol{D}_T^T||_2 \leq \frac{1}{\rho} O(\sqrt{\epsilon_0}), \tag{15}$$

*where $\epsilon_0 = O(\sqrt{\frac{\log n}{n}})$. Then, we have*

$$|x_v^T \boldsymbol{W}_I^* \boldsymbol{W}_T^* x_t^{\hat{y}} - \frac{1}{\rho} x_v^T \boldsymbol{D}_I \begin{bmatrix} 1 + \sigma_{inv}^2 & 2p_{spu} - 1 \\ 2p_{spu} - 1 & 1 + \sigma_{spu}^2 \end{bmatrix} \boldsymbol{D}_T^T x_t^{\hat{y}}||_2 \leq ||x_v||||x_t^{\hat{y}}|| \frac{1}{\rho} O(\sqrt{\epsilon_0}) \quad \leq \frac{1}{\rho} O(\sqrt{\epsilon_0} \log n). \tag{16}$$

In Lemma A.2 , we notice that

$$x_v^T \boldsymbol{D}_I \begin{bmatrix} 1 + \sigma_{inv}^2 & 2p_{spu} - 1 \\ 2p_{spu} - 1 & 1 + \sigma_{spu}^2 \end{bmatrix} \boldsymbol{D}_T^T x_t^{\hat{y}} = \hat{y}((y + \xi_1)(1 + \sigma_{inv}^2) + (a + \xi_2)(2p_{spu} - 1). \tag{17}$$

For inference, the model discriminates based on the similarity between the two modalities. Consider the scenario where the model encounters samples with spurious correlations ( $y = 1, a = -1$ ), and we apply the mask $\mathbf{M} = \begin{bmatrix} 1 & 0 \\ 0 & M \end{bmatrix}$ on vision modality. When CLIP makes an incorrect prediction, we have

$$\mathbf{M} x_v^T \boldsymbol{W}_I^* \boldsymbol{W}_T^* x_t^{\hat{y}=1} < \mathbf{M} x_v^T \boldsymbol{W}_I^* \boldsymbol{W}_T^* x_t^{\hat{y}=-1} \tag{18}$$

Note that $M$ follows a Bernoulli distribution, so $\mathbf{M} x_v = \begin{bmatrix} y + \xi_1 \\ p_{mask}(a + \xi_2) + (1 - p_{mask})\xi_0 \end{bmatrix}$ where $\xi_0$ can be considered as $\mathcal{N}(0, 0)$. The entire spurious part can be considered as following a Gaussian mixture distribution.

By substituting $\mathbf{M} x_v$ for $x_v$ in Eq. 16, we still obtain:

$$\frac{1}{\rho} \mathbf{M} x_v^T \boldsymbol{D}_I \begin{bmatrix} 1 + \sigma_{inv}^2 & 2p_{spu} - 1 \\ 2p_{spu} - 1 & 1 + \sigma_{spu}^2 \end{bmatrix} \boldsymbol{D}_T^T x_t^{\hat{y}=1} + \frac{1}{\rho} O(\sqrt{\epsilon_0} \log n) <$$
$$\frac{1}{\rho} \mathbf{M} x_v^T \boldsymbol{D}_I \begin{bmatrix} 1 + \sigma_{inv}^2 & 2p_{spu} - 1 \\ 2p_{spu} - 1 & 1 + \sigma_{spu}^2 \end{bmatrix} \boldsymbol{D}_T^T x_t^{\hat{y}=-1} - \frac{1}{\rho} O(\sqrt{\epsilon_0} \log n), \tag{19}$$

with Eq. 17 plugged in, denote $\epsilon_1 = O(\sqrt{\epsilon_0} \log n)$, we further have

$$2\left[(1 + \xi_1)(1 + \sigma_{inv}^2) + p_{mask}(-1 + \xi_2)(2p_{spu} - 1) - \epsilon_1\right] < 0. \tag{20}$$

Since $\xi_1(1 + \sigma_{inv}^2) + \xi_2(2p_{spu} - 1)p_{mask}$ is a Gaussian variable follows the distribution of

$$\xi_1(1 + \sigma_{inv}^2) + \xi_2(2p_{spu} - 1) \sim \mathcal{N}(0, (1 + \sigma_{inv}^2)^2\sigma_{inv}^2 + (2p_{spu} - 1)^2 p_{mask}^2 \sigma_{spu}^2),$$

then, we have

$$\begin{aligned}
&\Pr(2\left[(1 + \xi_1)(1 + \sigma_{inv}^2) + p_{mask}(-1 + \xi_2)(2p_{spu} - 1) - \epsilon_1\right] < 0) \\
&= \Pr_{v \sim \mathcal{N}(0,1)}(v < \frac{p_{mask}(2p_{spu} - 1) - 1 - \sigma_{inv} + \epsilon_1}{\sqrt{(1 + \sigma_{inv}^2)^2\sigma_{inv}^2 + (2p_{spu} - 1)^2 p_{mask}^2 \sigma_{spu}^2}}) \\
&= \Phi(\frac{p_{mask}(2p_{spu} - 1) - 1 - \sigma_{inv} + \epsilon_1}{\sqrt{(1 + \sigma_{inv}^2)^2\sigma_{inv}^2 + (2p_{spu} - 1)^2 p_{mask}^2 \sigma_{spu}^2}}),
\end{aligned} \tag{21}$$

where $\Phi$ is the CDF of the standard Gaussian distribution. Then, it suffices to know that the $\text{Err}_{y=1,a=-1}$ is lower bounded by $\Phi(k) - o(1)$ and upper bounded by $\Phi(k) + o(1)$ where $k = \frac{p_{mask}(2p_{spu} - 1) - 1 - \sigma_{inv}}{\sqrt{(1 + \sigma_{inv}^2)^2\sigma_{inv}^2 + (2p_{spu} - 1)^2 p_{mask}^2 \sigma_{spu}^2}}$ which also applies to the case $y = -1, a = 1$.

Next we demonstrate the impact of $p_{mask}$. $\frac{dk}{dp_{mask}} = \frac{(2p_{spu} - 1)(1 + \sigma_{inv}^2)^2\sigma_{inv}^2 + (1 + \sigma_{inv}^2)(2p_{spu} - 1)^2\sigma_{spu}^2 p_{mask}}{((1 + \sigma_{inv}^2)^2\sigma_{inv}^2 + (2p_{spu} - 1)^2\sigma_{spu}^2 p_{mask}^2)^{3/2}}$ Considering $1/2 < p_{spu} < 1$ and $0 \le p_{mask} \le 1$, $k$ increases monotonically as $p$ increases. Notice that $1 - p_{mask}$ is the actual mask ratio and $p_{mask} = 1$ represents the vanilla error bound without mask. As the proportion of the masked spurious components increases, the model's error bound gradually decreases.

## B. Experimental Details

All of our experiments are conducted on a single NVIDIA GeForce RTX 4090 GPU. The class prompts and auxiliary prompts for ERICT used in experiments are shown in Table 5.

Table 5: Summary of prompts in main experiments.

| Dataset | Class Prompts | Auxiliary Prompt for ERICT |
|---|---|---|
| Waterbirds | " a photo of a landbird " 
 " a photo of a waterbird " | " bird in photo " |
| CelebA | " a photo of a person with non-blonde hair " 
 " a photo of a person with blonde hair " | " hair in photo " |
| Urbancars | " a photo of an urban car " 
 " a photo of a country car " | " car in photo " |

## C. Pseudocode

---

**Algorithm 1:** Step 1 of ERICT-C

---

**Input** : Image $x_v$, Image encoder $f_v$, Text encoder $f_t$, Class prompts $x_t$, Temperature parameter $\tau$, Top-K strategy parameter $k$, Patch number $p$.

`// Auxiliary Prompt`

$z_v^{cls} = \frac{f_v(x_v)[cls]}{||f_v(x_v)[cls]||}$ ;

**for** $x_t^i \in x_t$ **do**

$\quad z_t^i = \frac{f_t(x_t^i)}{||f_t(x_t^i)||}$ ;

$\quad S_i = z_v^{cls} \times z_t^i$ ;

**end**

$S \leftarrow \text{sort}(S, \text{descending})$ ;

$S_k \leftarrow S[1:k]$ ;

$\hat{z}_t \leftarrow z_t$ corresponding to $S_k$ ;

$\hat{z}_t^a \leftarrow \frac{1}{k} \sum_{i=1}^{k} \hat{z}_t[i]$ ;

`// Get Mask`

$z_v^{patches} = \frac{f_v(x_v)[patch]}{||f_v(x_v)[patch]||}$ ;

**for** $i = 1$ **to** $p$ **do**

$\quad S_i = z_v^{patches}[i] \times \hat{z}_t^a$ ;

**end**

$S \leftarrow \text{sort}(-1 \times S, \text{descending})$ ;

$S \leftarrow \text{sort}(S, \text{descending})$ ;

$\rho \leftarrow \max \left( i : S_i/\tau > \frac{1}{i} \sum_{i=1}^{i} S_i/\tau \right)$ ;

$l \leftarrow \frac{1}{\rho} \left( \sum_{i=1}^{\rho} S_i/\tau - 1 \right)$ ;

**for** $i = 1$ **to** $p$ **do**

$\quad$ **if** $S_i > l$ **then**

$\quad\quad M_i \leftarrow 1$ ;

$\quad$ **end**

$\quad$ **else**

$\quad\quad M_i \leftarrow 0$ ;

$\quad$ **end**

**end**

**Output** : $M$

---

---

**Algorithm 2:** Step 2 of ERICT-C

---

**Input** : Image Token sequence $X$, Image encoder $f_v$, Text encoder $f_t$, Class prompts $x_t$, Corresponding mask $M$.

**for** $BLK \in f_v[:-2]$ **do**
  $\quad | \quad X \leftarrow BLK(X)$ ;
**end**
**for** $BLK \in f_v[-2:]$ **do**
  $\quad | \quad Q, K, V \leftarrow \text{Proj}_{QKV}(X)$ ;
  $\quad | \quad A_{cls} = \text{softmax}(M \cdot \frac{Q_{cls}K^T}{\sqrt{d_k}})$ ;
  $\quad | \quad A_{patches} = \text{softmax}(\frac{Q_{patches}K^T}{\sqrt{d_k}})$ ;
  $\quad | \quad X = X + \text{Proj}([A_{cls}, A_{patches}] \cdot V)$ ;
  $\quad | \quad X = X + \text{FFN}(X)$ ;
**end**
$z_v = \text{Proj}(X)$ ;
**for** $x_t^i \in x_t$ **do**
  $\quad | \quad z_t^i = \frac{f_t(x_t^i)}{||f_t(x_t^i)||}$ ;
**end**
$\hat{y} = \text{argmax}_{z_t^i \in z_t}(z_v \times z_t^i)$ ;
**Output** : $\hat{y}$

---

## D. More results

### D.1. Imagenet Series Datasets

In these section, we present the results of ERICT-C on multi-category datasets. Imagenet (Deng et al., 2009) is a widely used large-scale vision dataset containing more than 14 million images covering 1,000 categories. ImageNet-A (Hendrycks et al., 2021b) is a subset of ImageNet consisting of hard samples that were misclassified by the ResNet model, which contains 200 categories. ImageNet-R (Hendrycks et al., 2021a) is constructed by stylizing some images in the ImageNet in various ways, aiming to test the robustness of the model when facing style-transferred images. It contains 200 categories.

We present the performance of ERICT-C with the top-3 strategy as an example across three datasets in Table 6. As shown in table 6, our method outperforms vanilla CLIP on all these datasets, with particularly significant improvements observed on ViT-L14.

Table 6: Experimental results of ERIT* on Imagenet dataset with top-3 strategy.

| | Imagenet | | | Imagenet-A | | | Imagenet-R | | |
|---|---|---|---|---|---|---|---|---|---|
| | ViT-B/32 | ViT-B/16 | ViT-L/14 | ViT-B/32 | ViT-B/16 | ViT-L/14 | ViT-B/32 | ViT-B/16 | ViT-L/14 |
| ZSCLIP | 59.31 | 64.01 | 70.21 | 29.07 | 46.32 | 66.75 | 66.54 | 73.58 | 85.19 |
| ERICT-C | **59.45**↑0.14 | **64.02**↑0.01 | **72.15**↑1.94 | **31.19**↑2.12 | **48.19**↑1.87 | **70.03**↑3.28 | **67.25**↑0.71 | **75.23**↑1.65 | **87.19**↑2.00 |

### D.2. Visualization Experiments

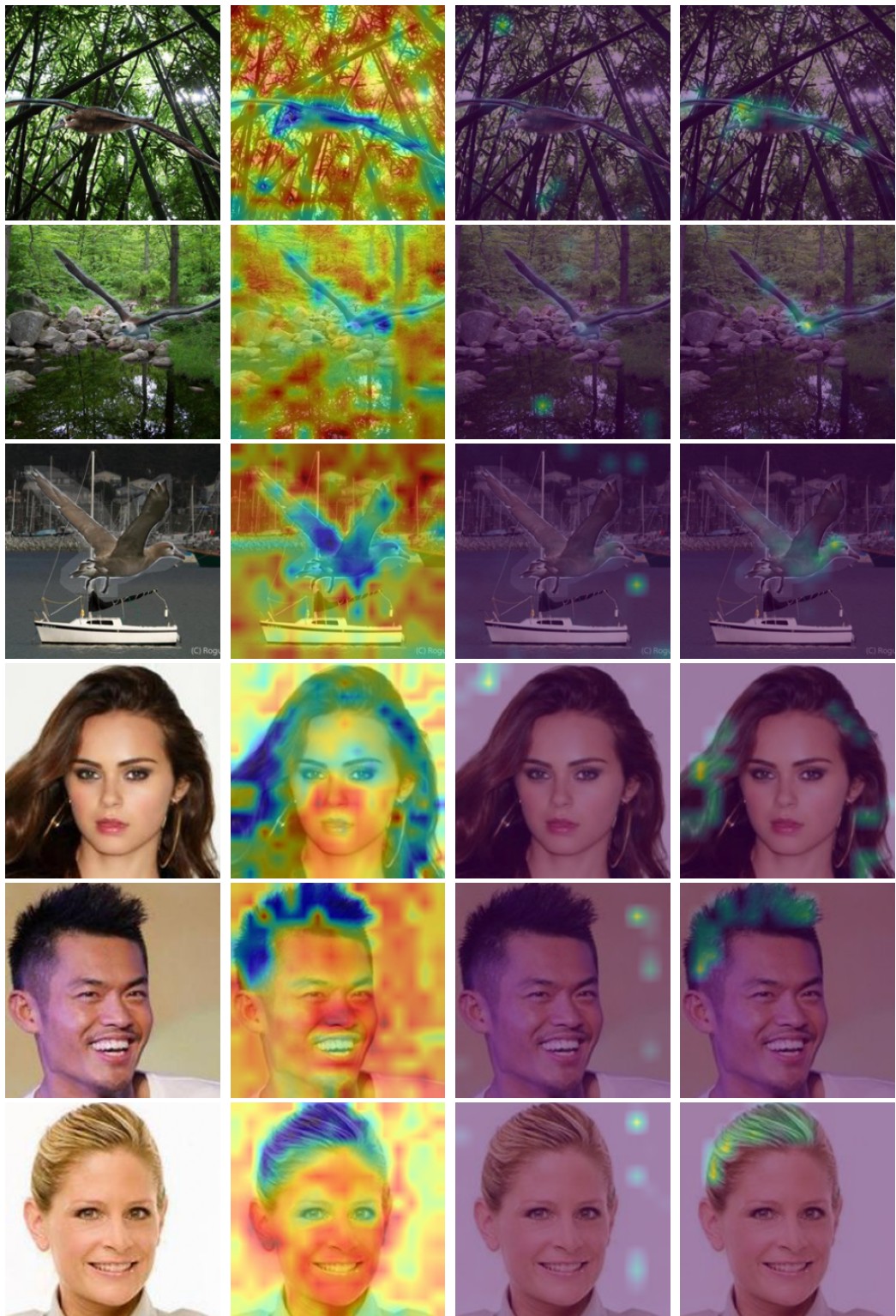

Figure 7: CAM comparison diagram. From left to right, the sequence is as follows: the original image, the score heatmap visualization ( to identify the desired tokens), the original CAM image, and the CAM image with ERICT.

