# OpenReview forum: "ERICT: Enhancing Robustness by Identifying Concept Tokens in Zero-Shot Vision Language Models"
_ICML.cc/2025/Conference — ICML 2025 poster_

### Official Review · Reviewer_Cu15 · 2025-02-18

**Overall Recommendation:** 3

**Summary:**

This paper introduces ERICT, a novel method to enhance model robustness by identifying concept tokens and mitigating spurious correlations at the inference stage. ERICT operates in two key steps: (1) identifying invariant concept tokens using auxiliary prompts to generate a token-level mask and (2) applying the mask to the CLS token's attention weights in the vision encoder, ensuring the model focuses on relevant image regions.

**Claims And Evidence:**

No, there are several unclear aspects that need further clarification:

(i) The paper observes that “tokens whose semantics align more closely with the prompt tend to have lower similarity scores,” but provides only limited examples to support this claim. Given that this observation appears counterintuitive, stronger empirical validation is necessary.

(ii) Based on this observation, the method should ideally select the top-k lowest similarity token embeddings. However, it seems to do the opposite, which requires further explanation.

(iii) The selection of auxiliary prompts appears to require domain knowledge, and ERICT-C is only evaluated using the top-3 version. Both aspects require a more in-depth investigation.

**Essential References Not Discussed:**

No.

**Experimental Designs Or Analyses:**

Yes. The main experiments should be fine. However, some ablation studies are missed. For example, how to select a better auxiliary prompt and ERICT-C with diverse top-k.

**Methods And Evaluation Criteria:**

Yes. The proposed method is pretty straightforward and should be easily used to enhance the zero-shot performance of VLMs.

**Other Comments Or Suggestions:**

No.

**Other Strengths And Weaknesses:**

Strengths:

+The proposed method is simple and elegant.

+The experiments on selected datasets show the effectiveness of the method.



Weaknesses:

- Several unclear aspects exist in this work as mentioned before.

- Compared with baseline method, ROBOSHOT, the method missed experiments on other datasets, e.g., PACS, VLCS and CXR14.

**Questions For Authors:**

My questions have been mainly mentioned above.

**Relation To Broader Scientific Literature:**

Compared with existing works, this work focuses on zero-shot learning without the help of extra LLMs and labeled data. However, expert knowledge to design the auxiliary prompt is needed, which might not be trivial for all tasks. The alternative method, ERICT-C, is not well evaluated.

**Theoretical Claims:**

Yes, there is a simple proof to support the assumption, which is correct.

---

> ### Author Rebuttal · Authors · 2025-04-01
>
> > **Q1: The paper observes that “tokens whose semantics align more closely with the prompt tend to have lower similarity scores,” but provides only limited examples to support this claim. Given that this observation appears counterintuitive, stronger empirical validation is necessary.**
>
> **A1**: Thanks for the reviewer’s suggestion. We have provided additional image results (https://anonymous.4open.science/r/rebuttal-6618/more-image.pdf) from different datasets to further support this point. In each row, the leftmost image is the original image, followed by the score heatmaps and score maps for three different concepts in the image. These results further validate our findings.
>
> > **Q2: Based on this observation, the method should ideally select the top-k lowest similarity token embeddings. However, it seems to do the opposite, which requires further explanation.**
>
> **A2**: We thank the reviewer for pointing out this error. In Equation (6), $S_i$ should actually be $S_{i}^{'}$, with $S^{'} = Sort(-1 \times S)$. In the implementation, consistent with the finding in line 178, we use sparsification to select tokens with lower scores to obtain the final visual representation. We will correct this error in the revised version. Additionally, we would like to re-explain the top-k strategy, which is used in ERICT-C to select potential class prompts for aggregation as auxiliary embeddings. In this process, we sort the inference results from vanilla CLIP and select the class prompts that are most similar to the image.
>
> > **Q3: The selection of auxiliary prompts appears to require domain knowledge, and ERICT-C is only evaluated using the top-3 version. Both aspects require a more in-depth investigation.**
>
> **A3**: ERICT uses the superclass of the task category as prior knowledge to compute the auxiliary embedding. This prior does not require domain experts and is not complex. It is also used in the baseline method, PerceptionCLIP.
>
> ERICT-C, on the other hand, does not rely on any prior knowledge. It computes embeddings for each class prompt in the classification task and aggregates the top-k embeddings with the highest similarity to obtain the auxiliary embedding. To demonstrate the effectiveness of the top-k strategy, we presented results with top-3 in the paper on multi-category datasets such as ImageNet-R. Additionally, to further illustrate this, we have provided more results for different values of k.
>
> |   ViT-L/14   | Imagenet-1k | Imagenet-A | Imagenet-R|
> | :-: | :-: | :-: | :-: |
> | K=2  |    72.32    |   69.95    |   87.30 |
> | K=3  |    72.15    |   70.03    |   87.19    |
> | K=5  |    72.03    |   69.76    |   86.99    |
>
> > **Q4 Compared with existing works, this work focuses on zero-shot learning without the help of extra LLMs and labeled data. However, expert knowledge to design the auxiliary prompt is needed, which might not be trivial for all tasks. The alternative method, ERICT-C, is not well evaluated.**
>
> **A4**: ERICT does require the introduction of certain priors to use auxiliary prompts, but these priors are similar to the simple assumption of "bird in photo" in the waterbirds dataset, which is also used in the widely discussed work PerceptionCLIP in this field. At the same time, ERICT-C does not rely on any priors, addressing this limitation and demonstrating superior performance. We have also provided additional experiments (e.g., Tables 1, 2, 3, 6 in our manuscript and the table in Q5) to validate the effectiveness of our method.
>
> > **Q5: Compared with baseline method, ROBOSHOT, the method missed experiments on other datasets, e.g., PACS, VLCS and CXR14.**
>
> **A5**: We sincerely clarify that our paper primarily focuses on the issue of spurious correlations, while PACS, VLCS, and CXR14 are domain generalization (DG) benchmarks, not spurious correlation problems. Following the reviewer's suggestion and the experimental setup of ROBOSHOT, we conducted a comprehensive comparison on PACS, VLCS, and CXR14. The experimental results show that our method still demonstrates strong robustness on these DG benchmarks.
>
> |ViT-L/14||PACS|||VLCS|||CXR14(BiomedCLIP)||
> |:-:|:-:|:-:|:-:|:-:|:-:|:-:|:-:|:-:|:-:|
> |Model|WG|AVG|Gap|WG|AVG|Gap|WG|AVG|Gap|
> |ZSCLIP|79.8| 98.1 | 18.3 | 4.2| 72.6 | 68.4 | 28.9 | 55.3  |26.4|
> |ROBOSHOT|83.9 | 98.1 | 14.2 | 12.6 | 71.1 | 58.5 | 41.6 | 56.2  | 14.6 |
> |ERICT|83.1| 96.4 | 13.3 | 39.8 | 73.2 | 33.4 | 42.3 | 56.7  | 14.4 |
> |ERICT-C|83.3| 97.2 | 13.9 | 44.3 | 78.9 | 34.6 | 45.6 | 57.2  | 11.6 |

---

> > ### Comment · Reviewer_Cu15 · 2025-04-03
> >
> > Thanks for the authors' response, which has solved most of my concerns. I will increase my score and please add these responses to the revision.

---

> > > ### Author Response · Authors · 2025-04-03
> > >
> > > We are glad to know that your concerns have been effectively addressed. We are very grateful for your constructive comments and questions, which help improve the clarity and quality of our paper. Thanks again!

---

### Official Review · Reviewer_89KE · 2025-03-09

**Overall Recommendation:** 4

**Summary:**

This paper presents ERICT, a novel method to enhance the robustness of vision-language models (VLMs) by mitigating spurious correlations at the inference stage. The approach identifies concept tokens to create a token-level mask, which is then applied to the vision encoder’s attention mechanism. Experimental results demonstrate that ERICT improves overall and worst-case performance, achieving state-of-the-art results.

**Claims And Evidence:**

In this paper, the authors argue that while fine-tuning methods are somewhat effective in mitigating the spurious correlation problem, they come with additional computational costs and rely heavily on the quality of prompts, without fully leveraging the vision modality. To address these limitations, the authors propose the ERICT method, which enhances robustness without requiring additional training, assistance from LLMs, or access to group labels. To validate the effectiveness of the proposed approach, the authors employ an error probability bound of the model.

The paper lacks an in-depth discussion on the disentanglement of invariant and spurious factors in Section 5. While it is intuitively feasible to decompose image samples into invariant and spurious factors, a more detailed exploration of the disentanglement approach is necessary.

**Essential References Not Discussed:**

To the best of my knowledge, the paper includes imporant related works.

**Experimental Designs Or Analyses:**

I have reviewed the authors’ experimental design. They conducted experiments on three widely used datasets with spurious correlations and compared their method against several state-of-the-art approaches from multiple perspectives. Additionally, visualization experiments were designed to further demonstrate the robustness of the proposed method. However, the experiments on different prompts, as shown in Table 5, were not presented.

**Methods And Evaluation Criteria:**

The proposed method is generally reasonable and has been validated for effectiveness from both average performance and worst-case perspectives. Additionally, the authors have theoretically demonstrated that the method can reduce the model’s probabilistic error bound. However, there are still some aspects of the method that warrant further discussion.

**Other Comments Or Suggestions:**

The caption of Figure 2 should describe the specific functions of the two steps to help readers quickly understand the content of the figure.

**Other Strengths And Weaknesses:**

Strengths

- The proposed ERICT method mitigates the spurious correlation problem without the need for fine-tuning or additional complex prompts, reducing computational costs and dependency on prompt quality.

- Extensive experiments demonstrate that ERICT improves overall model performance, including that of the worst-performing group, and achieves new state-of-the-art results, highlighting its effectiveness in addressing the robustness challenges of vision-language models.

Weaknesses

- The authors introduce a threshold $l$ in Equation (5), but do not further analyze it in the experimental section. It is recommended that the authors include two parts in the ablation study: (1) the impact of different threshold values on the experimental results; (2) the effect of different search threshold methods on performance.

- In line 189, the authors mention that the auxiliary prompt is very important, but they do not further analyze why the superclass of the task category is better than the ground-truth category. Additionally, are there other forms of prompts worth further discussion and investigation?

**Questions For Authors:**

- The authors should further clarify the rationale behind using a binary mask to mitigate spurious correlation issues in VLMs. Specifically, they should elaborate on how the binary mask influences feature selection, information filtering, and the model’s final decision-making process to enhance the understanding and persuasiveness of the proposed approach.

- Pre-trained VLMs typically exhibit strong generalization capabilities but can still be affected by spurious correlations. What are the underlying reasons for this?

**Relation To Broader Scientific Literature:**

The key contribution of this paper lies in proposing a novel method to mitigate the spurious correlation problem in vision-language models. By directly applying a concept token mask during the inference stage, this method enhances the model’s robustness and generalization ability, overcoming the limitations of existing methods that rely on large-scale language models and labeled data. Therefore, it represents a novel contribution to the literature.

**Theoretical Claims:**

The authors validate the effectiveness of the proposed binary mask approach by analyzing the model’s error probability bound. A thorough examination of the theoretical analysis in Section 5 and the corresponding appendix confirms its correctness.

---

> ### Author Rebuttal · Authors · 2025-04-01
>
> > **Q1: Section 5 lacks a detailed discussion on disentangling invariant and spurious factors, requiring further exploration.**
>
> **A1**: Consistent with previous works[1-2], our theoretical analysis adopts the classic data assumption, which disentangles spurious datasets into invariant and spurious parts. In this paper, our method identifies concept tokens to effectively extract both invariant and spurious information.
>
> [1] Robust Learning with Progressive Data Expansion Against Spurious Correlation.NeurIPS 2023.
>
> [2] A Sober Look at the Robustness of CLIPs to Spurious Features. NeurIPS 2024.
>
>
> > **Q2: The experiments on different prompts were not presented.**
>
> **A2**: We thank the reviewer for pointing out this issue. The experiments in our paper used the commonly adopted prompt template "a photo of a {}". Additionally, we have conducted extensive experiments on CelebA to evaluate the performance of ERICT under different prompt templates, where  ``Prompt 80`` represents the average of 80 templates designed by OpenAI for CLIP.
>
> |ViT-L/14|Prompt template|WG|AVG|Gap|
> |:-:|:-:|:-:|:-:|:-:|
> |ERICT|a photo of a {CLASS}|85.00|86.43|1.43|
> |ERICT|Prompt 80|84.51|86.36|1.85|
> |ERICT|a {CLASS}|84.89|86.80|1.91|
> |ERICT-C|a photo of a {CLASS}|85.44|86.53|1.09|
> |ERICT-C|Prompt 80|84.48|86.16|1.69|
> |ERICT-C|a {CLASS}|85.00|87.81|2.81|
>
> > **Q3: The authors introduce a threshold l in Eq (5), but don't analyze its impact or the effect of different threshold methods.**
>
> **A3**: We further clarify the threshold ``l``. In this paper, ``l`` is dynamically calculated based on the sparsification process, considering that the number of invariant information tokens varies across different images. Each image in the task has a different threshold depending on the variation in spurious information (see Equations (4) and (5)). The key parameter influencing the sparsification process is the temperature coefficient, which we have already analyzed in the ablation study section (lines 385-420). To further demonstrate the effectiveness of this process, we provide comparative experiments on different threshold settings using the CelebA.
>
> |ViT-L/14||WG|AVG|Gap|
> |:-:|:-:|:-:|:-:|:-:|
> |ERICT|sparse|**85.00**|86.43|**1.43**|
> |ERICT|50%|76.67|88.01|11.34|
> |ERICT|80%|79.44|88.82|9.38|
> |ERICT-C|sparse|**85.44**|86.53|**1.09**|
> |ERICT-C|50%|78.33|87.76|9.43|
> |ERICT-C|80%|84.44|86.24|1.80|
>
> > **Q4: The discussion about the auxiliary prompt. Other worthy prompt forms.**
>
> **A4**: We sincerely apologize for not clearly expressing this point in the submitted paper. What we actually want to convey is that the auxiliary embedding is very important as it guides model inference. Specifically, ERICT utilizes superclass information, while ERICT-C employs a Top-K strategy to aggregate class prompt embeddings. Since they obtain auxiliary embeddings in different ways, their performance varies across different tasks. We do not want to emphasize that the "superclass of the task category is better than the ground-truth category", but to point out the limitations of the auxiliary prompt (which requires certain priors), and then introduce ERICT-C.  We will clarify this issue in the final version.
>
> > **Q5: The caption of Figure 2.**
>
> **A5**: Thanks for the reviewer's suggestion. Figure 2 mainly includes two steps: In Step 1, we construct an auxiliary embedding $z_t^a$ to identify tokens containing invariant information, obtaining a token-level mask. In Step 2, we apply this token-level mask within the attention mechanism of the vision encoder, making tokens containing spurious information "invisible" to the [CLS] token.
>
> > **Q6: The authors should clarify how the binary mask mitigates spurious correlations, particularly its impact on feature selection, information filtering, and decision-making.**
>
> **A6**: Thanks for the interesting question raised by the reviewer. According to our findings (line 178), the non-CLS tokens in the visual encoder capture local information. The original visual representation is obtained through the attention interaction mechanism between the CLS token and other tokens. By using a binary mask, we "shield" the tokens that focus on spurious information from the CLS token, so that the final representation no longer attends to spurious-related information (as demonstrated clearly in our T-SNE visualization). This prevents the model from making decisions based on spurious information.
>
> > **Q7: What are the underlying reasons for spurious correlation in VLMs?**
>
> **A7**: Thanks for your interesting question. In our view, although pre-trained VLMs use large-scale image-text pairs, they still fundamentally follow a supervised learning paradigm, which cannot fully avoid the influence of fundamental learning issues, a point supported by related work [3]. Unfortunately, real-world data are full of statistical biases.
>
> [3] Mitigating Spurious Correlations in Multi-modal Models during Fine-tuning. ICML 2023

---

> > ### Comment · Reviewer_89KE · 2025-04-07
> >
> > Decided to raise my score according to the thorough rebuttal, which solves most of my questions.

---

> > > ### Author Response · Authors · 2025-04-07
> > >
> > > We are pleased to learn that your questions have been effectively addressed. Thank you sincerely for your constructive feedback, which has significantly improved the clarity and quality of our paper. Thanks again!

---

### Official Review · Reviewer_USPN · 2025-03-13

**Overall Recommendation:** 3

**Summary:**

The paper introduces ERICT and ERICT-C, mitigate spurious correlations in vision-language models (VLMs) during zero-shot inference. These approaches aim to enhance model robustness by identifying invariant features within image tokens and focusing the model's attention on relevant regions through token masking. The methods are theoretically grounded and demonstrate significant improvements in performance.

## update after rebuttal

The rebuttal resolves my concerns, I am inclined to accept this paper.

**Claims And Evidence:**

The authors' claims regarding improved robustness and reduced spurious correlations are supported by experimental results. However, the distinction between ERICT and ERICT-C needs clarification, and the underperformance of ERICT-C in some scenarios requires further investigation.

**Essential References Not Discussed:**

The paper could benefit from experiments on more data such as MetaShift, and Living-17 datasets, as well as refering more debiasing algorithm like [1][2]

[1] Debiased Fine-Tuning for Vision-language Models by Prompt Regularization
[2] CLIPood: Generalizing CLIP to Out-of-Distributions

**Experimental Designs Or Analyses:**

The experimental designs are robust, with evaluations on multiple datasets and backbones. However, the choice of the top-3 strategy in ERICT-C lacks justification, and the limitations mentioned in the text are not adequately addressed in subsequent sections.

**Methods And Evaluation Criteria:**

The proposed methods aim to address spurious correlations by leveraging invariant features in image tokens. However, the distinction between ERICT and ERICT-C is not clearly articulated. The evaluation criteria (WG, AVG, Gap) are appropriate for assessing robustness.

**Other Comments Or Suggestions:**

The paper presents a novel and effective approach to enhancing robustness in VLMs. To strengthen the work, the authors should:
1.	Clarify the distinction between ERICT and ERICT-C.
2.	Investigate why ERICT-C underperforms in certain scenarios.
3.	Provide justification for the top-3 strategy choice.
4.	Address the limitations mentioned in the text.
5.	Include results on additional datasets and compare with more algorithms.

**Other Strengths And Weaknesses:**

Strengths:
•	The approach is efficient as it does not require additional training or group labels.
•	The paper provides visualizations and ablation studies that help understand the method's effectiveness.
Weaknesses:
•	The distinction between ERICT and ERICT-C is not clearly explained.
•	ERICT-C underperforms ERICT in some cases as shown in Table 1, 2, 3, suggesting limitations in the auxiliary prompt strategy.
•	The choice of the top-3 strategy in ERICT-C lacks experimental justification.
•	The paper does not compare with ERM[1], AFR[2], and CFR[3] algorithms.
[1]Principles of risk minimization for learning theory.
[2]Simple and fast group robustness by automatic feature reweighting.
[3]Calibrating multimodal representations: A pursuit of group robustness without annotations.

**Questions For Authors:**

see weaknesses

**Relation To Broader Scientific Literature:**

The work contributes to the field of robust vision-language models by proposing a novel approach to mitigate spurious correlations without requiring additional training data or group labels.

**Theoretical Claims:**

The error probability bound theorem supports the effectiveness of the approach.

---

> ### Author Rebuttal · Authors · 2025-04-01
>
> > **Q1: However, the distinction between ERICT and ERICT-C needs clarification**
>
> **A1**: The key difference between ERICT and ERICT-C lies in the way they obtain the auxiliary embeddings. As shown in Step 1 of Figure 2, ERICT uses auxiliary prompts (e.g., "bird in photo") as input to the text encoder to obtain auxiliary embeddings. In contrast, ERICT-C constructs prompts using class names (e.g., "a photo of a landbird") as input to the text encoder. Then, ERICT-C ranks the embeddings output by the class names based on similarity and aggregates the top-K most similar embeddings to obtain the auxiliary embedding, where K is a hyperparameter less than or equal to the maximum number of categories in the dataset.
>
> > **Q2: The underperformance of ERICT-C in some scenarios requires further investigation.**
>
> **A2**: ERICT and ERICT-C each have their strengths and weaknesses under different settings, primarily due to their use of two different strategies (as shown in Q1) for obtaining auxiliary embeddings. The difference in how these embeddings are obtained leads to varying performance in the task.
>
>
> > **Q3: The choice of the top-3 strategy in ERICT-C lacks justification and the limitations mentioned in the text are not adequately addressed in subsequent sections.**
>
> **A3**: We apologize for any confusion caused to the reviewer. We have clarified the Top-K strategy. In the experiments, the value of k should be less than or equal to the number of categories in the dataset. For example, for the binary classification task on the Waterbirds dataset, K=2. Similarly, we have provided experiments with different values of K on three distorted versions of the Imagenet dataset, as shown below.
>
> The limitation mentioned in the paper mainly refers to the suboptimality of the auxiliary prompt. In fact, we discuss this limitation in the "Impact of the auxiliary prompt" section of the ablation study (lines 400-412). How to obtain an auxiliary embedding with the best guiding significance is indeed a meaningful topic for further exploration. However, the main contribution of this paper lies in enhancing the model's robustness during the inference phase without relying on training, the assistance of LLMs, or group labels. We hope to address this limitation for future work.
>
> |  ViT-L/14 |Imagenet-1k |Imagenet-A|Imagenet-R|
> | :-: | :-: | :-: | :-: |
> |K=2|72.32|69.95|87.30|
> |K=3|72.15| 70.03| 87.19|
> |K=5|72.03| 69.76| 86.99|
>
> > **Q4: The paper could benefit from experiments on more data such as MetaShift, and Living-17 datasets, as well as referring to more debiasing algorithms.**
>
> **A4**: As suggested by the reviewer, we add extensive experiments comparing the MetaShift and Living-17 datasets. The experimental results are shown below. These experiments demonstrate the robustness of our method across different datasets. Please note that the code for "Debiased Fine-Tuning for Vision-Language Models by Prompt Regularization" mentioned has not been open-sourced. The experiments for CLIPood and additional baseline results are presented in Q5.
>
> |    ViT-B/16     |              | MetaShift |             |             | Living-17 |             |
> | :-: | :-: | :-: | :-: | :-: | :-: | :-: |
> |  model  |      wg      |    avg    |     gap     |     wg      |    avg    | gap         |
> | ZSCLIP  |    87.56     |   94.96   |     7.4     |    31.5     |   93.7    | 62.2        |
> |  ERICT  |  **91.89**   |   95.38   |  **3.49**   | 38.2 |   94.1    | 55.9 |
> | ERICT-C | 90.27 |   95.33   | 5.06 |  **38.5**   |   94.2    | **55.7**    |
>
> > **Q5: The paper does not compare with ERM, AFR and CFR algorithms.**
>
> **A5**: We thank the reviewer for pointing out this issue. We have added detailed experiments comparing our method with extensive baseline methods. It is worth noting that although these baseline methods are training-based, our approach is training-free. However, in certain settings, our method outperforms these training-based methods, further demonstrating the robustness of our approach.
>
> |   ViT-L/14      |          |       | Waterbirds |       |       | CelebA |      |
> | :-: | :-: | :-: | :-: | :-: | :-: | :-: | :-: |
> |         | Training | Wg    |    Avg     | Gap   | Wg    | Avg    | Gap  |
> | ERM     | Yes      | 57.9  |    97.6    | 39.7  | 30.4  | 94.6   | 64.2 |
> | AFR     | Yes      | 73.4  |    88.2    | 14.8  | 70.0  | 85.2   | 15.2 |
> | CFR     | Yes      | 88.2  |    96.8    | 8.6   | 84.8  | 87.8   | 3.0  |
> | CLIPood |   Yes    | 79.6 |    94.1    | 14.5 | 31.1 |  95.3  | 64.2 |
> |  ERICT  |    NO    | 61.2 |    74.1    | 12.9 | 85.0 |  88.6  | 3.6  |
> | ERICT-C | NO | 64.8 | 79.4 | 14.6 | 83.3 | 89.0 | 5.7 |

---

### Official Review · Reviewer_ZMVi · 2025-03-13

**Overall Recommendation:** 4

**Summary:**

This paper proposes ERICT, a zero-shot method to improve robustness in vision-language models by identifying “concept tokens” that represent invariant image features. ERICT uses auxiliary prompts to generate masks applied to attention weights, aiming to reduce spurious correlations. The authors evaluate ERICT on standard benchmarks (Waterbirds, CelebA, Urbancars), claiming substantial improvements, especially on worst-group accuracy.

**Claims And Evidence:**

The claims made in the paper are largely supported by thorough experimental results on multiple benchmarks (Waterbirds, CelebA, Urbancars), showing clear improvements in worst-group accuracy. Visualizations provided further illustrate that ERICT effectively shifts attention away from spurious features.

**Essential References Not Discussed:**

N/A

**Experimental Designs Or Analyses:**

The experimental design is sound and clearly demonstrates improved robustness.

**Methods And Evaluation Criteria:**

Yes, the proposed method directly addresses the spurious correlation issue in zero-shot VLMs, and the benchmarks used (Waterbirds, CelebA, Urbancars) are appropriate, clearly capturing improvements in robustness and worst-group accuracy.

**Other Comments Or Suggestions:**

N/A

**Other Strengths And Weaknesses:**

N/A

**Questions For Authors:**

N/A

**Relation To Broader Scientific Literature:**

The paper builds on existing literature addressing spurious correlations in VLMs, offering a zero-shot solution distinct from previous fine-tuning or LLM-based methods.

**Theoretical Claims:**

Theorem 5.2 establishes the theoretical error bound after applying the proposed mask. The theorem itself is well-formulated, and the intuition (reducing spurious correlations lowers the error probability) is reasonable.

---

> ### Author Rebuttal · Authors · 2025-03-31
>
> Thank you for your positive feedback on our work. We **sincerely appreciate your recognition of our contributions and the significance of the research problems we address**. Your support further strengthens our confidence in the proposed approach. If you have any additional questions or suggestions, we would be happy to discuss them.

---

### Decision · Program_Chairs · 2025-05-01

**Decision:**

Accept (poster)

**Comment:**

This paper received consistently positive reviews. The initial concerns were primarily related to the clarity of the experimental setup and implementation details. The rebuttal effectively addressed these issues.

As a result, the AC concurs with the reviewers and recommend acceptance. It is encouraged to incorporate the clarifications provided in the rebuttal into the camera-ready version to further strengthen the paper.